# The Use of Physical Activity in Coping with Test Anxiety Among Students: A Systematic Review

**DOI:** 10.3390/ijerph22121786

**Published:** 2025-11-26

**Authors:** Simone C. D. Amorim, João Paulo Langsdorff-Serafim, Alberto Souza Sá Filho, Luís Vicente Franco Oliveira, Viviane Soares, Claudia Santos Oliveira, Rodrigo Franco Oliveira, Raphael Martins Cunha, Júlio B. Mello, Iransé Oliveira-Silva

**Affiliations:** 1Graduate Program in Human Movement and Rehabilitation, Evangelical University of Goiás (UniEvangélica), Anápolis 75083-515, GO, Brazil; simone.amorim@seduc.go.gov.br (S.C.D.A.); alberto.filho@unievangelica.edu.br (A.S.S.F.);; 2eFidac Research Group, Escuela de Educación Física, Pontificia Universidad Católica de Valparaíso, Valparaíso 2362807, Chile; julio.mello@pucv.cl

**Keywords:** physical activity, exercises, anxiety, test anxiety, mental health, students

## Abstract

Test anxiety is a specific form of anxiety that occurs in evaluative situations, comprising psychological, physiological, and behavioral reactions associated with concern about outcomes. Aiming to analyze the relationship between physical activity and coping with anxiety, particularly test anxiety, in students aged 13 to 25 years, as well as the effects on mental health, this review followed the PRISMA guidelines. We searched PubMed, Web of Science, Lilacs, Cochrane, Medline, and Scielo databases, from January 2014 to January 2024, to select randomized and controlled clinical studies (including cluster-randomized) and quasi-experimental studies in English and Portuguese, focusing on the relationship between physical activity and anxiety, particularly test anxiety, in the context of mental health. Two independent researchers performed data extraction. The quality of the studies was assessed using the PEDro scale, and the risk of bias was evaluated using the RoB 2.0 tool. A total of 537 articles were found, and 14 were included via Rayyan. The selected scientific studies indicate an effective relationship between physical activity and coping with general anxiety, but clinical studies specifically related to test anxiety are still scarce. The reviewed studies support this positive relationship, which could be further explored in future research and diversified practices offered to students.

## 1. Introduction

Test anxiety is a specific form of anxiety that occurs in evaluative or testing situations, involving psychological, physiological, and behavioral reactions associated with concerns about outcomes, and is understood as synonymous with the fear of failure, generating tension and worry [1,2]. It is a response of an organism to stress, extensively studied since 1950, and negatively affects academic performance as well as students’ mental health [1,2,3,4,5,6]. Tests have been widely used in educational environments to evaluate students’ knowledge [7,8], and test anxiety can occur before, during, or after assessments, impacting students’ lives [1,2,9,10]. Due to test anxiety, students may have difficulty interpreting, evaluating, and organizing their thoughts to respond to test items [11].

Although it shares some characteristics with other types of anxiety, such as generalized or social anxiety, which can occur in a variety of everyday contexts, test anxiety has particularities that differentiate it from other types [12]. Physical symptoms such as sweating, tachycardia, and muscle tension are related to the affective dimension known as emotionality, which refers to physiological arousal. In addition to these, students experience specific intrusive thoughts, such as the fear of failing or not meeting expectations, a cognitive dimension known as worry [13,14,15]. These thoughts are less common in other types of anxiety and directly interfere with academic or professional performance [15]. Students experiencing this type of anxiety may have difficulties concentrating or recalling information during assessments, as well as a tendency to procrastinate in the weeks and months leading up to exams, limiting their study capacity [13,16]. Although both types of anxiety share risk factors, such as genetic predisposition and life experiences, test anxiety may be more related to social pressures and performance expectations in educational environments [1,13,17,18]. Test anxiety is a specific form of anxiety that presents unique characteristics compared to other forms, such as generalized or social anxiety [15].

Test anxiety has consistently shown a negative relationship with educational outcomes, as tests have taken on a much more prominent role in a multitude of important educational decisions [12]. Currently, there is much greater pressure for results, and many students feel overwhelmed by stress [19]. It is noted that since 2000, there has been a growing prevalence of mental illnesses, including test anxiety, partly due to increased attention to overall mental health, with more refined diagnoses and greater treatment options, and partly due to a real increase in the number of affected youths [19]. Anxiety and depression have risen among university students, especially after the outbreak of the coronavirus disease 2019 (COVID-19) [20]. The pandemic has led to complex situations of social isolation and prolonged screen exposure among adolescents [21]. Physical activity has shown promise in addressing mental health issues, with regular physical activity aiding in anxiety reduction and improving psychological well-being [22,23]. However, the specific relationship between physical activity and test anxiety is less explored, especially in this context where stress related to academic performance may have intensified in recent years. Thus, it is necessary to analyze this relationship during the period from 2014 to 2024.

Test anxiety, as a mental health issue, represents a concern for adolescents [24,25] because it extends into adulthood and constitutes an increasing global health problem [26]. The World Health Organization (2021) reported that among the 13% of young people with mental disorders, 3.6% of those aged 10 to 14 years and 4.6% of those aged 15 to 19 years suffer from some form of anxiety disorder, often evolving into depressive cases [27], generally starting between the ages of 12 and 25 [28]. The Programme for International Student Assessment (PISA) highlighted that Brazil ranks second among countries where students feel the most anxious during tests [29].

In the school environment, little attention is given to anxiety related to examinations; however, both academic performance and subjective well-being play a fundamental role in people’s lives [30,31]. Therefore, interventions targeting this type of anxiety can serve as a practical strategy to guide preventive efforts [32], since anxiety affects students’ evaluation of their own skills and achievements [33], making this study relevant for understanding and addressing this phenomenon, especially during adolescence, a period characterized by intense cognitive, emotional, and social changes [34].

Studies show that anxiety can be triggered by stress, one of the main causes of changes in autonomic nervous system (ANS) function [35]. In response to stressors, the physiological response of an organism involves generalized excitatory activity throughout the body and brain, increasing activity in the amygdala [36], activation of the hypothalamic–pituitary–adrenal (HPA) axis, and the release of adrenaline in the initial phase of stress and cortisol, which attempts to promote the organism’s resistance to the stressor(s) through the rapid mobilization of energy substrates.

These chemical regulators promote an adaptive reaction in search of homeostasis; however, if stressors persist and cortisol levels remain elevated, the organism may suffer harmful effects, such as increased respiratory rate, blood pressure, skin conductance, muscle tension, and heart rate in sympathetic fight-or-flight responses [37,38,39], detectable through heart rate variability (HRV), one of the promising markers of autonomic balance [38,40,41].

Physical activity (PA), defined by the American College of Sports Medicine (ACSM) [42,43] as any bodily movement that results in energy expenditure, with exercise representing a planned, structured, and repetitive subgroup of PA, is considered in its broad scope in this study. Physical activity encompasses a wide range of body movements, including those used as adjuncts to therapeutic practices, such as cognitive behavioral therapy (CBT). These non-pharmacological approaches are recognized for their ability to influence mental and physical health [44,45], integrating various physical activities that range from breathing exercises to more elaborate practices like yoga [46] and qigong [47]. These types of physical activity provide both physiological and psychological benefits and are essential, as they promote mindfulness, body awareness, and emotional regulation—key characteristics for treating anxiety and depression [45,46,47]. These complex interactions [48] can enhance the understanding of the use of physical activity in coping with anxiety, especially test anxiety among students.

Regular PA has been referred to as a factor that increases vagal tone [24] and may play a role in stress and anxiety regulation, as it promotes physiological changes and adaptations in the body, influencing the sympathetic nervous system and the reactivity of the HPA axis by altering the release of corticotropin-releasing factor (CRF) from the hypothalamus and adrenocorticotropic hormone (ACTH) from the anterior pituitary, modulating stress and anxiety reactivity in humans [49].

In this regard, studies confirm that PA through aerobic exercises and relaxation techniques helps reduce test anxiety [50,51] and provides positive effects on physical and mental health, being used to alleviate mental illnesses [52,53,54,55]. They also show that individuals with appropriate levels of PA have a lower risk of mental health disorders compared to those with lower levels of PA [42,56,57], and interventions focused on impulse control and resistance to temptation (healthy habits), such as participation in sports, can be beneficial for managing symptoms of irritability, anxiety, and depression [58,59].

Studies also highlight the relationship between physical activity and anxiety in children and adolescents [60,61], but they show a limitation of research in this area. Although several studies address anxiety in its different contexts [51,62,63,64,65] and physical activity is increasingly used to address a variety of issues, including anxiety, there is still no consensus on the dose–response relationship or which types of physical activity [66,67] are needed to mitigate anxiety, which continues to grow among young people. Therefore, it is essential to understand the relationship between physical activity and coping with anxiety in contemporary times, especially test anxiety, which significantly affects students’ academic performance.

When it comes to anxiety, particularly test anxiety, there is a lack of studies that examine the extent to which interventions are effective in improving students’ mental health. To fill this gap, the relationship between physical activity and test anxiety needs to be better understood and examined. Considering that physical activity is often the first step in lifestyle modification for the prevention and management of chronic diseases [49], and considering other studies [6,53,61,64,68,69,70,71] and their contributions, this study focused on research conducted in the last ten years (2014–2024). Through a systematic review, it explored potential strategies for coping with anxiety, especially test anxiety.

This review is necessary to confirm assertions made in previous research and aims to analyze the relationship between physical activity and coping with anxiety, particularly test anxiety, in students aged 13 to 25, as well as assessing its effects on mental health.

## 2. Materials and Methods

The researchers reviewed studies that analyzed the relationship between physical activity (PA) and anxiety, specifically test anxiety, in the context of mental health. For the methodology, the researchers followed the Preferred Reporting Items for Systematic Reviews and Meta-Analyses (PRISMA) guidelines [72].

The review protocol was retrospectively registered in the Open Science Framework (OSF) under the following DOI: https://doi.org/10.17605/OSF.IO/WVD5U.

### 2.1. Eligibility Criteria

The PICOS approach was adopted to determine the inclusion criteria [72], which are as follows: (P) including students aged 13 to 25 years, covering educational stages from secondary to higher education and equivalent qualifications; (I) addressing PA or exercise; (C) comparing between those who participated and those who did not; (O) the outcome was anxiety, specifically test anxiety and mental health; and (S) randomized and controlled trials, cluster-randomized trials, and quasi-experimental studies published between January 2014 and January 2024. Peer-reviewed publications written in English or Portuguese, regardless of the country, were considered.

Studies were excluded if they met any of the following criteria: studies not relating PA to anxiety, test anxiety, or mental health; studies where the subjects were not students; studies outside the selected time frame; and studies written in languages other than English or Portuguese. Duplicate articles were removed using the Rayyan application [73].

### 2.2. Search Strategy

The databases PubMed, Web of Science, Lilacs, Cochrane, and Medline were consulted from January 2014 to January 2024, with the goal of including studies on the relationship between physical activity (PA) and the management of anxiety, specifically test anxiety, in the context of mental health. The MESH terms used were “physical activity”, “exercise”, “anxiety”, “test anxiety”, “mental health”, and “students”. The terms were combined as follows: “physical activity” OR “exercise” AND “anxiety” OR “test anxiety” AND “mental health” AND “students”, and this combination was employed across all databases.

### 2.3. Selection and Data Extraction Procedure

The screening of titles and abstracts was conducted based on keyword searches available through scientific databases on the web. Eligible studies were retrieved, read, and evaluated. Articles were extracted using the Rayyan application [73]. Two researchers independently conducted the screening, and in cases of discrepancies, a third researcher was consulted. Data on the selected studies included author, year, and reference; the number in the experimental and control groups; age; intervention, duration, and total number (n); PA and anxiety (effects); measures, anxiety outcomes, and results (Table 1). Detailed characterization of the interventions was carried out using the TIDieR (Template for Intervention Description and Replication) checklist, aiming for completeness and replicability of the information [74].

### 2.4. Methodological Quality Evaluation

The quality of the selected studies was independently assessed by two researchers (SA, JP) using the PEDro scale, which ranges from 0 to 10 points. The first criterion of the scale was excluded as it assesses the external validity of the study. The quality was considered high if the score was between 7 and 10, intermediate if the score was between 5 and 6, and low if the score was between 0 and 4 [75]. Discrepancies were resolved by consensus, and the results are presented in Table 2.

### 2.5. Bias Risk Evaluation

To assess the risk of bias, the RoB2 tool from the Collaboration Network was used, considering five areas: randomization, deviation from interventions, missing outcome data, outcome measures, and selection of reported outcomes. The quality levels were classified as low risk, intermediate risk (some concerns), and high risk of bias [76].

**Table 1 ijerph-22-01786-t001:** The characteristics of studies which analyzed the relationship between physical activity and anxiety, especially test anxiety, in the context of mental health.

	Author, Year [Ref]	Ctl (n)	Exp (n)	Age	Intervention	Pa and Anxiety	Med	Out Tde	Results
1	Kamath et al., 2017 [77]	15	15	MD = 21 years old	Alternate nostril breathing exercise. Duration: 15 min (*n* = 30).	There was no significant effect on test anxiety in the ANB group (14.17 ± 4.78 and 9.3 ± 6.85) compared to C (14.51 ± 5.08 and 14.61 ± 7.75; *p* = 0.852). Only a potential anxiolytic effect in stressful situations.	VAMS SSPS	TA ANB 0.765	ANB did not show a significant reduction in anxiety induced by public speaking simulation, only lower anxiety scores on the VAMS, indicating a potential anxiolytic effect of ANB exercise in acute stressful situations.
2	Baghurst et al., 2014 [78]	132	SM = 124 CF = 131 PA = 144	MD = 21 years old	SM = cognitive behavioral exercises; practice mental and physical relaxation. PA = sports and games. CF = aerobic and anaerobic exercises. Duration: 3 days a week for 50 min for 16 weeks (*n* = 531).	Regarding anxiety, the PA group (25.9 ± 5.8; *p* > 0.05) and CV (28.2 ± 6.9; *p* > 0.05) showed lower means than SM (33.9 ± 5.2) and C (31.6 ± 7.2) under stress conditions. The PA group (29.6 ± 6.4 and 22.1 ± 3.1; *p* < 0.05), compared to the beginning and end of the semester, showed a significant reduction in test anxiety, as did SM (39.4 ± 5.8 and 28.5 ± 6.4; *p* < 0.05). CF showed no changes in anxiety (28.1 ± 6.2 and 28.3 ± 5.7).	TAS	TA SM 1.599 CF 0.102 PA 1.354	SM and PA significantly reduced test anxiety, perceived stress, and personal exhaustion in university students.
3	Eather et al., 2019 [79]	26	Uni-HIIT = 27	MD = 21 years old	Uni-HIIT = aerobic and resistance exercises. Duration: three HIIT sessions per week for 8 weeks, with sessions lasting 8 to 12 min each, with a rest interval ratio of 30:30 s (*n* = 53).	There were no significant effects of Uni-HIIT on anxiety in C (13.9 ± 1.47 and 14.20 ± 1.83) and Uni-HIIT (14.29 ± 1.50 and 14.39 ± 2.37; *p* = 0.709). Significant group-by-time effects on improvement in cardiorespiratory fitness (*p* = 0.004) and muscular fitness (*p* = 0.006) were observed.	STAI	GA HIIT 0.173	The Uni-HIIT program showed significant improvements in cardiorespiratory fitness and muscular fitness. No significant effects were found for anxiety.
4	Parker et al., 2016 [80]	PS = 86 AC = 89	PA = 88 PST = 85	MD = 20 years old	PA = behavioral activation, with PA chosen by the participant. PST, PA, and AC versus PS and PST, and PS and AC. Duration: 6 weekly sessions (*n* = 174).	With physical activity, there was a reduction in anxiety scores on the BAI (F (2121) = 109.43, *p* < 0.001), a reduction in depression scores on the MADRS (F (2125.4) = 79.52, *p* < 0.001), and a significant reduction in the BDI-II (F (2122) = 97.22, *p* < 0.001).	BAI BDI-II MADRS	GA PS/PST 0.301 PA/AC 0.471	PA using a behavioral activation approach helped reduce anxiety scores and symptoms of depression in school-aged youth compared to a psychoeducational intervention.
5	Zheng et al., 2021 [81]	485	469	MD = 13 years old	AF = eye exercises and relaxation. Duration: 10 min of AF; 4 times a day for 2 weeks (*n* = 954).	There was a significant reduction in anxiety (β = −0.66 (−1.04, −0.27), *p* = 0.001) with the physical activity intervention aiming to achieve behavioral change.	SCAS	GA	The digital behavioral change intervention reduced anxiety and visual fatigue among 7th-grade students during online schooling associated with COVID-19.
6	Zhang et al., 2023 [82]	39	39	MD = 19 years old	BAD Baduanjin (Qigong) Duration: 3 days a week, 1 h a day for 12 weeks (*n* = 78).	Regarding anxiety, there was a significant reduction in the scores of psychological symptoms and emotional distress (SCL90) in BAD (0.74 ± 0.58 and 0.58 ± 0.45) and C (0.76 ± 0.49 and 0.73 ± 0.38; *p* < 0.05). There was a significant reduction in weight and BMI, increased vital capacity, and improvement in blood pressure compared to C.	SCL90	GA BAD 0.395	The study indicates that the practice of Baduanjin can reduce anxiety and benefit the physical and mental health of university students.
7	Papp et al., 2019 [83]	27	27	MD = 25 years old	High-intensity hatha yoga. Duration: 1 h, once a week for 6 weeks (*n* = 54).	Regarding anxiety, the study did not show consistent effects of HIY 6.9 ± 2.9 and 7.4 ± 4.2; *p* = 0.50) compared to C (7.6 ± 4.2 and 7.0 ± 4.3; *p* = 0.36).	HADS	GA 0.288	After the 6-week intervention, there were no differences between the groups in terms of anxiety, depression, stress, sleep, or self-rated health.
8	Murray et al., 2022 [84]	29	48	MD = 23 years old	Aerobic resistance exercises. Duration: 30 min classes twice a week, for 8 weeks (*n* = 77).	Decrease in initial and post-test anxiety scores in WeA (7152 ± 5517 and 6652 ± 5225) and in WeM (7786 ± 4475 and 6429 ± 4710), although not statistically significant.	GAD-7	GA 0.145	The study suggests that aerobic resistance exercises, as well as mindfulness exercises, may be beneficial in alleviating anxiety related to academic stress in university students.
9	Bentley et al., 2022 [85]	18	SP-1= 6 SP-2= 7 GP= 12	MD = 17 years old	Breathing exercises (slow, diaphragmatic, and prolonged exhalation breathing). Duration: 5 min for 5 weeks.	There were no significant changes in anxiety in SP1 (38.9 ± 9.2 and 39.3 ± 8.2; *p* = 1.00), SP2 (38.0 ± 9.7 and 35.0 ± 13.9; *p* = 0.99), GP (38.9 ± 10.3 and 37.4 ± 12.1; *p* = 0.99), and C (38 ± 10.6 and 36.8 ± 12.4).	STAI	GA SP-1 0.128 SP-2 0.141 GP 0.037	There were no significant changes as a result of the breathing exercises regarding anxiety (*p* > 0.05 for all).
10	Rosenberg et al., 2021 [86]	5	SDBE = 8 BIOF = 9	MD = 25 years old	Self-directed breathing exercises and use of a biofeedback device. Duration: 3 times a day for three minutes over 3 weeks (*n* = 34).	There was a significant reduction in anxiety in BIOF (2.76 ± 0.67 and 2.13 ± 0.59; *p* < 0.05). In SDBE (2.84 ± 0.56 and 2.69 ± 0.51) and C (2.8 ± 0.61 and 2.65 ± 0.59), no significant reductions in anxiety were observed.	TAI QMP	TA SDBE 0.005 BIOF 0.820	University students who used the biofeedback device showed a significant reduction in test anxiety symptoms.
11	Gallego et al., 2014 [87]	42	PE= 42 MBCT = 41	MD = 20 years old	Mindfulness and PE = physical education classes. Duration: 8 sessions once a week for 1 h (*n* = 125).	There was no significant reduction in anxiety for mindfulness (4.47 ± 3.78 and 3.46 ± 2.41; *p* = 0.480 and d = 0.318) and for physical education (5.79 ± 4.26 and 5.09 ± 5.05; *p* = 0.418 and d = 0.149), only a small effect. For both, there was a reduction in stress levels (*p* < 0.05).	DASS-21	GA PE 0.161 MBCT 0.342	University students who participated in the intervention had a significant reduction in stress levels and average levels of anxiety and depression compared to the control group, although this was not significant.
12	Tasan et al., 2021 [88]	70	PB = 70	MD = 19 years old	Pranayamic breathing exercises (*n* = 140).	There was a significant reduction in learning anxiety in PB (86.1 ± 19.8 and 82.3 ± 13.5; *p* = 0.048) and in test anxiety (58.2 ± 12.1 and 55.4 ± 13.4; *p* = 0.023), as well as improvement in listening and reading comprehension skills in English after PB practice (*p* > 0.05).	FLTAS	GA 0.093	Pranayama breathing exercises significantly helped reduce levels of learning anxiety and test anxiety in undergraduate university students studying English.
13	Cho et al., 2016 [89]	12	MBP = 12 CRP = 12	MD = 20 years old	Conscious breathing exercises and cognitive reappraisal practice. Duration: 6 days of individual training before and 30 min daily for 1 day for 7 days (*n* = 36).	The intervention with MBP (49.75 ± 5.71 and 38.58 ± 10.04; *p* < 0.001) and PRC (50.58 ± 6.26 and 41.17 ± 8.94; *p* < 0.001) resulted in a significant reduction in test anxiety. In the MBP group, there was also an increase in positive automatic thoughts.	RTA	GA MBP 0.780 CRP 0.626	Both mindful breathing and cognitive reappraisal practices were effective in reducing test anxiety in undergraduate students.
14	Li et al., 2021 [57]	13	14	MD = 22 years old	Resistance training with an intensity of 70% 1RM. Duration: 8 weeks, training twice a week, 72 h interval between sessions, 40 min training, 5 min relaxation (*n* = 27).	With the TR intervention, there was a significant reduction in anxiety, observed in the SAS (58.1 ± 6.2 and 39.0 ± 4.2; *p* = 0.005) and in the parameters of heart rate variability: SDNN (*p* = 0.012); HF (*p* = 0.018); and LF/HF (*p* = 0.047). Additionally, there was an increase in muscular strength (*p* < 0.05).	SAS	GA TR 1.543	The TR intervention significantly reduced anxiety levels in students, significantly increased SDNN and HF, and decreased the LF/HF ratio, indicating an improvement in autonomic nervous system function.

Abbreviations: CTL = control; EXP = experimental; C = control group; MD = average; MED = measures of anxiety or depression; OUT = primary and secondary outcome; TA = test anxiety; GA = general anxiety; TDE = effect size; BMI = body mass index; BP = blood pressure. Intervention: ANB = alternative nostril breathing exercise; SM = cognitive behavioral stress management; PA = physical activity; CF = cardiovascular fitness; Uni-HIIT = aerobic and resistance exercises; PST = problem solving therapy; PS = psychoeducation; AC = counseling; BAD = Baduanjin; HYH = high-intensity hatha yoga; WeA =aerobic resistance exercises; WeM = yoga mindfulness; ERAD = self-directed breathing exercises; BIOF = biofeedback; EF= physical education; CBT = cognitive behavioral therapy; PB = pranayamic breathing; MBP = mindful breathing practice; CRP = cognitive reappraisal; TR = resistance training; HRV = heart rate variability; GP = guided rhythm diaphragmatic breathing group; SP-1 = slow self-paced diaphragmatic breathing; SP-2 = group 2, slow diaphragmatic breathing at your own pace; CO2TT = carbon dioxide tolerance test. Measurements (test anxiety): VAMS = Visual Analogue Mood Scale; SSPS = Public Speaking Self-Statement Scale; TAS = Test Anxiety Survey; STAI = Spielberger State–Trait Anxiety Inventory; TAI = Test Anxiety Inventory; FLTAS = Foreign Language Test Anxiety Scale; RTA = The Revised Test Anxiety Scale. Measurements (anxiety): DASS-21 = Chinese version of the Depression, Anxiety and Stress Scale; BAI = Beck Anxiety Inventory; SCAS = Spence Children’s Anxiety Scale; SCL90 = Checklist-90 Symptom Scale; HADS = Hospital Anxiety and Depression Scale; GAD-7 = Spitzer Generalized Anxiety Disorder Scale; QPM = Perceived Change Questionnaire; SAS = Zung’s Self-rating Anxiety Scale; SCL90 = Psychological and psychiatric symptoms; BDI-II = Beck Depression Inventory-II; MADRS = Montgomery–Asberg Depression Rating Scale.

**Table 2 ijerph-22-01786-t002:** The PEDro scale: Assessing the methodological quality of articles that analyzed the relationship between physical activity and anxiety, especially test anxiety, in the context of mental health.

	Autor	1	2	3	4	5	6	7	8	9	10	11	Total
1	Kamath et al., 2017 [77]	X	X	X	X				X	X	X	X	7
2	Baghurst et al., 2014 ** [78]	X			X				X	X	X	X	5
3	Eather et al., 2019 [79]	X	X	X	X					X	X	X	6
4	Parker et al., 2016 [80]	X	X	X	X			X	X	X	X	X	8
5	Zheng et al., 2021 * [81]	X	X	X	X				X	X	X	X	7
6	Zhang et al., 2023 [82]	X	X	X	X	X	X	X	X	X	X	X	10
7	Papp et al., 2019 [83]	X	X	X	X	X	X	X		X	X	X	9
8	Murray et al., 2022 [84]	X	X	X	X	X			X	X	X	X	8
9	Bentley et al., 2022 * [85]	X	X		X			X		X	X	X	6
10	Rosenberg et al., 2021 [86]	X	X	X	X					X	X	X	6
11	Gallego et al., 2014 [87]	X	X	X	X					X	X	X	6
12	Tasan et al., 2021 [88]	X	X		X				X	X	X	X	6
13	Cho et al., 2016 [89]	X	X	X	X		X	X	X	X	X	X	9
14	Li et al., 2022 [57]	X	X	X	X				X	X	X	X	7
	Mean												7.14

Eligibility criteria; 2. subjects were randomly assigned to groups; 3. allocation concealment; 4. similar groups; 5. therapist blinding; 6. participant blinding; 7. assessor blinding; 8. key outcome measures obtained in >85% of participants; 9. participants received treatment or the control condition as allocated (intention to treat); 10. Results of intergroup statistical comparisons reported for at least one key outcome; 11. Precision measures as measures of variability for at least one key outcome. * Cluster-randomized and ** quasi-experimental interventions assessed using the TIDieR checklist.

## 3. Results

A total of 537 articles were identified, and 54 duplicates were removed during the identification phase. In the screening phase, 426 were excluded after reviewing titles and abstracts, and 43 were excluded for not meeting the inclusion criteria; finally, 14 were selected after a full-text review by the authors (Figure 1). The publications under analysis included 2.518 participants and provided synthesized and described information regarding the relationship between physical activity (PA) and anxiety, as well as test anxiety (Table 1). In total, five studies had test anxiety as the primary outcome, seven had general anxiety as the main outcome, and two considered general anxiety as a secondary outcome (Table 1).

The average score obtained in the methodological quality assessment of the studies using the PEDro scale was 7.14, indicating average quality. Of the articles, 57.14% had a high score, while 42.86% had an intermediate score (Table 2).

In addition to the above considerations, it is important to highlight the risk of bias analysis, which showed that 21.43% of the articles had a high risk of bias, 35.71% had some concerns, and 42.86% had a low risk. This type of study is prone to biases due to the specific characteristics of each study [75,76] (Figure 2).

Although 64.28% of the 14 selected studies demonstrated a reduction in anxiety through the use of physical activity (PA) modalities, such as aerobic exercise, mindfulness, games, relaxation, walking, Baduanjin (Qigong), conscious and slow breathing, self-directed breathing, biofeedback, pranayamic breathing, and resistance training, the analysis of the selected studies revealed wide variability in the magnitude of the effects of physical activity and psychophysiological interventions on anxiety. The effect sizes ranged from 0.005 to 1.543, covering null to very large effects.

Resistance training showed the largest effect (RT = 1.543) [57], followed by biofeedback combined with psychoeducation (BIOF = 0.820) [86], mindful breathing practice (MBP = 0.780), and cognitive reappraisal (CRP = 0.626) [89]. Smaller magnitudes were observed for mindfulness (MBCT = 0.342) and physical education practice (PE = 0.161) [87].

Self-directed interventions, such as autonomous breathing and pranayama, presented small or negligible effects (0.005–0.141) [84,85,88]. Overall, the findings indicate that structured and supervised protocols tend to produce more robust effects compared to unsupervised approaches. It is noteworthy that only 14.28% of the studies incorporated complementary measures such as heart rate variability (HRV) and biological and/or physiological analyses to confirm the effectiveness of physical activity.

In terms of mental health, the studies supported claims of improved sleep quality [83], stress control [86,87], increased psychological well-being [82,86], more positive thinking [89], and better autonomic control in response to emotions [57].

In summary, most clinical trials addressed anxiety in general [57,79,80,81,82,83,84,87], while five trials specifically examined test anxiety [77,78,86,88,89], highlighting a scarcity of clinical studies in this area. Additionally, 78.57% of the studies involved university students, while only 21.43% focused on students in primary and secondary education.

## 4. Discussion

This systematic review, as intended, analyzed and synthesized evidence from studies on the relationship between physical activity (PA) and anxiety, with a particular focus on test anxiety in the context of mental health. It is evident that various approaches and interventions have been employed, reflecting a significant interest in PA, especially concerning mental health, often referred to as a secondary epidemic [90].

Several studies address anxiety in different contexts [51,61,62,63,64,65,66,67,91], supporting the claims of Kandola et al. (2018), which indicate that physical activity-based interventions represent a new approach to the treatment of various mental health conditions [91], including anxiety, which continues to grow among young people [27,29].

In this sense, there are several important gaps in the literature to be explored in the context of mitigating anxiety, such as the underlying mechanisms of the effects of physical activity, ideal physical activity protocols, methods to improve adherence, the importance of physical fitness [91], and the dose–response relationship [66,67], areas that are being increasingly investigated in research.

In this systematic review assessing the effects of exercise interventions on students’ test anxiety, it was found that in a previous meta-analysis, exercise interventions effectively reduced test anxiety compared to the control group [6]. However, the limitation of excluding “non-physical” activities, such as muscle relaxation, meditation, mindfulness, and other interventions, raises concerns about limiting the scope of PA possibilities and potentially omitting some interventions. Additionally, another meta-analysis noted that PA alone might be effective in reducing anxiety. However, due to the relatively small sample size and the use of self-report questionnaires as the sole anxiety measure, a dose–response relationship for PA has still not been established and requires further study [92]. Thus, given the lack of studies expressing the dose–response relationship of PA with test anxiety and the effectiveness of interventions in improving students’ mental health (ages 13 to 25), this review sought to find explanations in the scientific literature to address this gap. Moreover, considering other studies [6,53,61,64,68,69,71,92,93], focused on the last ten years (2014–2024).

Many of the clinical trials on anxiety, particularly test anxiety, have focused on research with university students, revealing the increasing presence of anxiety. Studies suggest that the transition from adolescence to adulthood is marked by uncertainties and challenges that can exacerbate symptoms of anxiety and depression [94,95,96]. This period involves ongoing brain maturation, particularly in the limbic and cortical regions, which undoubtedly plays a role in physiological and emotional changes coinciding with adolescence [97]. The age range of 13 to 25 represents a significant transitional period where many young people face academic and professional stressors, increasing their vulnerability to anxiety and depression disorders. Other research shows that the roots of these feelings can develop during adolescence and extend to affect mental health in later stages [94,98,99]. However, in relation to the researched topic, only five trials were found directly related to test anxiety, indicating a scarcity of clinical studies. Therefore, the investigation in this age range is crucial for understanding how young people cope with anxiety, especially test anxiety, in different contexts.

This review included a variety of randomized controlled trials, cluster-randomized trials, and quasi-experimental studies related to managing anxiety in students through different interventions from various perspectives, among which physical activity is recognized for its significant influence on anxiety, addressing physiological, neurochemical, and cardiovascular aspects [65,100]. Stress-related hormones, such as cortisol, are regulated by regular exercise, which also affects essential neurotransmitters like endorphins and serotonin. Endorphins, known for promoting feelings of well-being, and serotonin, which is crucial for mood regulation, play an important role in reducing anxiety [100]. Additionally, physical activity improves cardiovascular responses, reducing blood pressure and heart rate in stressful situations, which are fundamental for cardiovascular health and the ability to cope with emotional challenges [101].

Regular exercise is also related to neuroplasticity, where physical activity promotes adaptive changes in the brain, improving brain health, providing a protective effect against anxiety, and optimizing cognitive function [94]. Thus, physical activity not only benefits the body’s physiology but also plays a critical role in mental health, contributing to stress regulation and improving quality of life. However, the type, intensity, duration, and frequency of exercise, as well as the conditions under which it should be performed to effectively reduce stress load for different individuals, need to be better understood and defined [102]. Furthermore, two specific vulnerability factors for anxiety—anxiety sensitivity and social anxiety—may influence the extent to which individuals can tolerate exercise interventions [100].

For instance, one study identified anxiety as a normal human emotion that alerts and empowers individuals to cope with perceived stressful situations. When excessive, anxiety destabilizes the individual and leads to dysfunctional states. This study found that alternate nostril breathing (ANB) could have physiological effects, such as reducing heart rate and blood pressure, as well as cognitive effects, such as improved focus and attention. It may also have potential anxiolytic effects in acute stressful situations, providing mental relief and promoting physical and mental balance. Although there was a trend toward lower test anxiety scores in the experimental group, the study did not achieve statistical significance [77].

Similarly, breathing exercises appear to reduce anxiety. A study that examined a simple behavioral intervention using a breathing tool (biofeedback) as exclusive therapy for test anxiety showed a significant reduction in anxiety symptoms [86]. Additionally, diaphragmatic breathing, tested during the pandemic through hybrid teaching, seems to be a simple and quick practice that could be effective in coping with anxiety, as suggested by the authors who tested this intervention in a large group of students [85], but further studies with a larger sample size are lacking.

Other studies highlight that qigong exercises relieve anxiety and reduce stress among healthy individuals. Qigong-based cognitive therapy promotes improvements in physical health, with positive effects in the treatment of depression and anxiety in Chinese individuals [82]. Similarly, pranayama breathing techniques from yoga were used as a positive psychological exercise to mitigate foreign language learning anxiety (FLLA) and test anxiety (TA) among English-speaking undergraduate students studying at a Turkish university, significantly altering FLLA and TA levels [88]. In contrast, high-intensity hatha yoga practice showed no positive association with improvements in anxiety but did improve depression and sleep among participants [83].

In the context of sports, a study examining the influence of core strengthening training on physical performance in university athletes practicing aerobic gymnastics found that 10 weeks of gymnastics training, with heart rate (HR) controlled between 130 and 150 bpm, twice a week for 90 min each session, resulted in improvements in physical fitness, sleep, diet, and mental health, including reduced anxiety among the athletes. This resistance training practice showed a significant increase in heart rate variability (HRV), indicating moderation of autonomic nervous system disturbances [57].

Similarly, in a university setting, 53 young participants engaged in Uni-HIIT, an 8-week high-intensity interval training program. Although the data for anxiety or perceived stress were not statistically significant (*p* > 0.05), improvements in cardiorespiratory and muscular fitness were observed [79]. In another study addressing perceived stress, test anxiety, and personal burnout, the intervention groups showed significant improvements in stress, test anxiety, and physical fitness [78].

During the pandemic, research focused on managing anxiety, especially due to social isolation. A digital intervention combining physical activity (PA), eye relaxation, and stretching for 10 min during break intervals, four times a day, resulted in reduced anxiety and visual fatigue among children [81]. In a similar technological vein, web-based aerobic resistance exercises (WeActive) and mindfulness yoga exercises (WeMindful) were implemented with 77 university students to assess the effects of these interventions on symptoms of depression and anxiety. The study concluded that while there was a reduction in depression and anxiety, the decrease was not highly significant [84].

Within this broad context, aiming to assess the efficacy of low-intensity interventions, a study utilized lifestyle psychoeducation and physical activity (PA) as behavioral activation. It demonstrated that this intervention significantly reduced symptoms of depression and anxiety in young individuals diagnosed according to the DSM-IV (Diagnostic and Statistical Manual of Mental Disorders, 4th ed.) [80].

Regarding test anxiety [77,78,88], in addition to these studies, another study indicated that both conscious breathing and cognitive reappraisal practices led to reductions in test anxiety, although the sample size was relatively small. This suggests the need for further research to build on these findings [89]. Similarly, a study highlights that tests serve as a primary tool for academic and professional decision-making and are frequently used to determine future career opportunities. It reports that 20 to 40 percent of university students experience test anxiety and that high levels of test anxiety affect performance and quality of life and decrease self-esteem and confidence, thus directly impacting mental health. The study emphasizes the significant reduction in test anxiety scores in the group practicing breathing exercises with a biofeedback device [*p* = 0.011], considering it an alternative method for controlling test anxiety [86].

Based on the available evidence, the fourteen selected studies demonstrate that a wide range of physical activity (PA) practices—such as breathing exercises, sports, walking, games, qigong, yoga, and mindfulness—are promising for managing anxiety. However, clinical trials present several challenges that can significantly impact the conduct and outcomes of research. Key difficulties include small sample sizes [77,79,82,83,86,87,89]; high variability in the results [78]; preparation and planning time [88]; the short duration of interventions [79,82]; the lack of objective anxiety measures, such as heart rate and blood pressure [77]; participant adherence issues [80,84,85]; implementation challenges due to hybrid learning [85]; limitations related to gender selection [57]; and the absence of blinding for evaluators [86] and participants.

These factors can compromise statistical power and the ability to detect significant differences between intervention and control groups, complicate data interpretation, and hinder the generalization of findings. This underscores the need for more robust statistical analyses to control these variables.

Additionally, the lack of objective anxiety measures, such as heart rate and blood pressure, can limit the ability to accurately assess the impact of interventions on participants’ anxiety. Insufficient adherence to study protocols by participants may introduce biases and reduce the internal validity of randomized controlled trials (RCTs). Failure to assess participants’ PA levels can influence results, and restricting the sample to a single gender may limit the generalizability of findings. The absence of blinding for evaluators and participants can introduce performance and detection biases, affecting result interpretation and study credibility [103].

However, understanding test anxiety and its characteristics is essential for developing effective interventions using physical activity to help individuals manage their anxiety in assessment situations. Physical activity can have differing effects on individuals suffering from test anxiety compared to those facing generalized anxiety. While both groups may benefit from physical activity, for those experiencing test anxiety, it is important that physical activity considers not only physiological and psychological aspects but also the context in which they are situated to help reduce the tension and nervousness associated with specific assessment situations.

The articles analyzing the relationship between physical activity and coping with test anxiety in students aged 13 to 25 showed that students can benefit from physical activity, helping them to cope with anxiety in general. The interventions included active breaks involving physical activity and eye relaxation [81], resistance training [57], and mindfulness [87]. The activities varied from 5 min to 2 h. In relation to test anxiety, the most effective types of physical activity were self-directed breathing exercises and those using biofeedback devices [86], pranayama breathing exercises [88], conscious breathing exercises and cognitive reappraisal practice [89], cognitive behavioral exercises combined with mental relaxation practice [78], and physical activity through sports and games [78], with activities varying from 3 min to 50 min (Table 1).

This review highlights that physical activity lasting up to 20 min can be an intervention option for test anxiety, complementing recent information [6], and that breathing exercises combined with cognitive therapies are significantly important in coping with test anxiety.

Despite the observations pointed out, it is worth noting that the limitations of the studies regarding general anxiety, due to heterogeneity, do not allow for a more robust synthesis of evidence. The variability is evident in the nature and intensity of the interventions, ranging from qigong Baduanjin practice [82] and breathing exercises to high-intensity interval training (HIIT) [79] and digital interventions promoting physical activity [80,81]. Session frequency and duration also varied widely, with durations from 10 min [81] and interventions up to 12 weeks [82], affecting the magnitude and persistence of the effects. Regarding application format, face-to-face and supervised approaches [79,80,82] and technology-mediated interventions [81] were implemented. The participants’ characteristics are equally diverse, including female university students [82], homeschooled youth [81], general university students [79], and young people with mild to moderate symptoms of depression and anxiety [80], each group presenting distinct potential moderating factors. Finally, the diversity of measurement instruments used, such as SCL90 [82], SCAS [81], STAI [79], and BAI [80], contributes to results that are not directly comparable, since each tool has varied sensitivity and specificity in capturing the various dimensions of the anxiety construct.

In studies on test anxiety, heterogeneity-related limitations include the nature of the intervention, which ranges from respiratory biofeedback [86] and mindful breathing [89] to pranayama techniques [88]. Dosage also differs significantly, with daily practices lasting a few minutes to extended weekly sessions, over periods ranging from weeks to months [86,88,89]. The application format includes autonomous interventions with remote support [86,89] as well as implementations integrated into supervised classroom environments [88]. The participants are mostly university students, although with different anxiety profiles and contexts [86,88,89]. Finally, the measurement instruments are varied, each with its own specificity for assessing test anxiety and related constructs (TAI, DASS-21, WHO-QOL-BREF, RTA, ATQ-P, PANAS, ELLAS, FLTAS).

The heterogeneity observed among the studies reflects the multifactorial nature of the relationship between physical activity and anxiety, marked by different intensities, durations, modalities, and degrees of supervision. The results indicate that structured and supervised protocols, especially those based on biofeedback, mindful breathing, and resistance training, present greater effect magnitudes, while self-directed or unsupervised interventions show minimal impact. This variability suggests that the effectiveness of physical activity on anxiety depends not only on the type of exercise but also on the quality of implementation and the psychophysiological support involved.

This multiplicity of approaches and intervening factors highlights the complex interconnection among studies involving physical activity and anxiety, as well as the influence of multiple physiological, psychological, and social factors that must be considered [65,94,98,100,104,105,106]. It is important to emphasize that, although this review focuses on test anxiety, most of the available evidence refers to general anxiety. Therefore, the extrapolation of findings should be made cautiously, acknowledging that the gap in the literature regarding specific interventions for test anxiety persists and warrants further investigation in future studies.

The subject of this review is a relatively recent field of study, especially regarding the context of test anxiety and the use of physical activity as a coping mechanism, and despite the scarcity of clinical studies, the diversity of physical activity applications demonstrates the pursuit of preventive methods to address anxiety, particularly test anxiety, and to improve students’ mental health.

## 5. Conclusions

This review provides evidence of the positive relationship between physical activity (PA) and anxiety, particularly test anxiety, indicating a reduction in anxiety and improvements in mental health related to physical activity. This review suggests that PA can be an effective intervention. However, the diversity of PA interventions and methodological approaches poses challenges for comparing studies and interpreting results. Therefore, there is a need for future clinical trials to adopt specific methodologies regarding physical activity and test anxiety so that their analyses can allow for better comparison and interpretation of the data.

Over the past decade, most selected clinical trials have addressed anxiety in general, with only five studies specifically focusing on test anxiety, highlighting a gap in clinical research. Additionally, while many studies focus on university students, there is a lack of research on test anxiety among primary and secondary school students. Given the variety of PA and the current social factors exacerbating anxiety among young people, future studies should focus on more clinical trials with preventive and effective interventions to reinforce this positive relationship. Further research should also explore PA levels and their influence on managing test anxiety.

## Figures and Tables

**Figure 1 ijerph-22-01786-f001:**
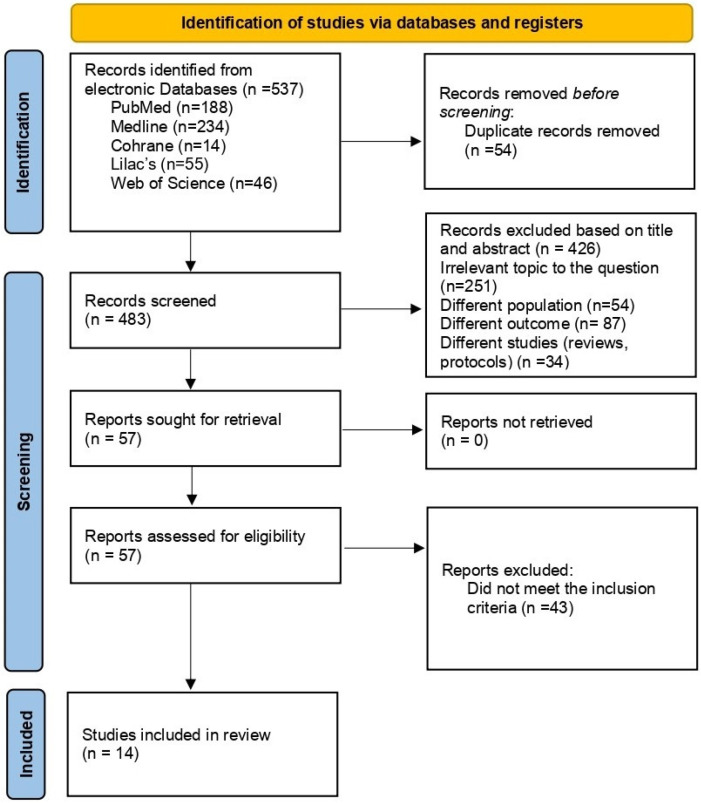
PRISMA 2020 flow diagram.

**Figure 2 ijerph-22-01786-f002:**
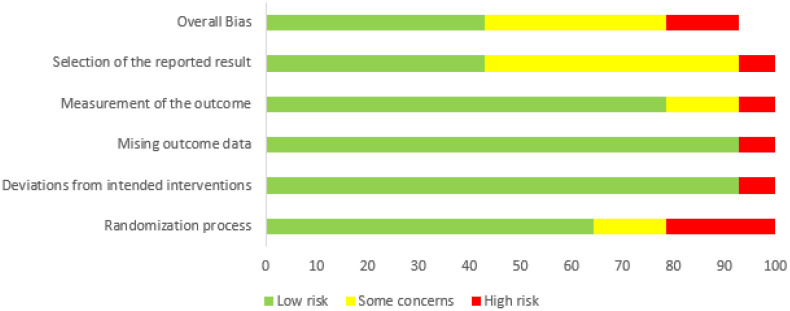
The risk of bias in articles that analyzed the relationship between physical activity and anxiety, especially test anxiety, in the context of mental health.

## Data Availability

No new data were created or analyzed in this study.

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
