# Peer review of "The Use of Physical Activity in Coping with Test Anxiety Among Students: A Systematic Review"

_ijerph, 2025, doi:10.3390/ijerph22121786_

Round 1

Reviewer 1 Report

Comments and Suggestions for Authors

Thank you for allowing me to review this timely and important topic. The manuscript addresses a high-impact question. However, the conceptual clarity, PRISMA‑level reporting, internal consistency, and language quality need substantial improvements.

The abstract asserts that the review “followed PRISMA” and included “randomized controlled clinical trials,” but the body suggests heterogeneous designs (e.g., cluster RCTs; possibly quasi‑experimental). Clarify design types in the abstract and avoid over‑claiming.

The PRISMA diagram  (figure 1):  the figure’s left pane labels “Records removed before screening” (duplicates 54; automated ineligible 402; total 426), whereas the text says 426 were excluded after reading titles/abstracts. Clarify at which stage these occurred and correct the labels accordingly. Please correct typos such as “Dublicates.

Please add a limitations paragraph on heterogeneity (interventions, outcomes, populations).

Some reference details appear incomplete or inconsistent (years, journal metadata, and links). Please verify all references and formatting (e.g., Gallego 2014 entry; Li et al. “2021/2022” inconsistency).

Comments on the Quality of English Language

Grammatical and typographic issues (e.g., comma decimals, capitalisation, and using the word “we” in a scientific context is incorrect; you can replace it with" the researchers") require professional English editing.  

Author Response

We would like to thank you and the editorial team for your time and constructive feedback, which greatly contributed to improving our manuscript.

Below, we provide detailed responses to each comment.

Sincerely,
Iransé Oliveira Silva, PhD

Detailed Responses to Reviewer 1's Comments and Suggestions

Comments 1: Thank you for allowing me to review this timely and important topic. The manuscript addresses a high-impact question. However, the conceptual clarity, PRISMA-level reporting, internal consistency, and language quality need substantial improvements.

Response 1: We greatly appreciate your feedback. We have completed all requested revisions and submitted the article to MDPI Author Services for English editing, aiming to improve our manuscript’s language.

Comments 2: The abstract asserts that the review "followed PRISMA" and included "randomized controlled clinical trials," but the body suggests heterogeneous designs (e.g., cluster RCTs; possibly quasi-experimental). Clarify design types in the abstract and avoid over-claiming.

Response 2: We agree with your observation and apologize for this oversight. We have amended the Abstract on page 1, lines 34, 35, and 36.

“[…]The databases PubMed, Web of Science, Lilac’s, Cochrane, Medline, and Scielo, from January 2014 to January 2024, allowed for the selection of randomized and controlled clinical studies (including cluster-randomized) and quasi-experimental studies in English and Portuguese, focusing on the[...]”

Comments 3: The PRISMA diagram (figure 1): the figure's left pane labels "Records removed before screening" (duplicates 54; automated ineligible 402; total 426), whereas the text says 426 were excluded after reading titles/abstracts. Clarify at which stage these occurred and correct the labels accordingly. Please correct typos such as "Dublicates."

Response 3: Thank you for your observation. The figure and text have been corrected with the appropriate values on page 5, lines 214-217.

P5… “A total of 537 articles were identified, and 54 duplicates were removed during the identification phase. In the screening phase, 426 were excluded after reviewing titles and abstracts, and 43 were excluded for not meeting the inclusion criteria; finally, 14 were selected after a full-text review by the authors (Figure 1)”.

Comments 4: Please add a limitations paragraph on heterogeneity (interventions, outcomes, populations).

Response 4: We appreciate your input. Text regarding heterogeneity (interventions, outcomes, populations) has been added on page 16, lines 433 to 479.

[…]Despite the observations pointed out, it is worth noting that the limitations of the studies regarding general anxiety, due to heterogeneity, do not allow for a more robust synthesis of evidence. The variability is evident in the nature and intensity of the interventions, ranging from qigong Baduanjin practice (82) and breathing exercises to high-intensity interval training (HIIT) (79) and digital interventions promoting physical activity (80,81). Session frequency and duration also varied widely, with durations from 10 minutes (81) and interventions up to 12 weeks (82), affecting the magnitude and persistence of the effects. Regarding application format, face-to-face and supervised approaches (79,80,82) and technology-mediated interventions (81) were implemented. The participants’ characteristics are equally diverse, including female university students (82), homeschooled youth (81), general university students (79), and young people with mild to moderate symptoms of depression and anxiety (80), each group presenting distinct potential moderating factors. Finally, the diversity of measurement instruments used, such as SCL90 (82), SCAS (81), STAI (79), and BAI (80), contributes to results that are not directly comparable, since each tool has varied sensitivity and specificity in capturing the various dimensions of the anxiety construct.

In studies on test anxiety, heterogeneity-related limitations include the nature of the intervention, which ranges from respiratory biofeedback (86) and mindful breathing (89) to pranayama techniques (88). Dosage also differs significantly, with daily practices lasting a few minutes to extended weekly sessions, over periods ranging from weeks to months (86,88,89). The application format includes autonomous interventions with remote support (86,89) as well as implementations integrated into supervised classroom environments (88). The participants are mostly university students, although with different anxiety profiles and contexts (86,88,89). Finally, the measurement instruments are varied, each with its own specificity for assessing test anxiety and related constructs (TAI, DASS-21, WHO-QOL-BREF, RTA, ATQ-P, PANAS, ELLAS, FLTAS).

The heterogeneity observed among the studies reflects the multifactorial nature of the relationship between physical activity and anxiety, marked by different intensities, durations, modalities, and degrees of supervision. The results indicate that structured and supervised protocols, especially those based on biofeedback, mindful breathing, and resistance training, present greater effect magnitudes, while self-directed or unsupervised interventions show minimal impact. This variability suggests that the effectiveness of physical activity on anxiety depends not only on the type of exercise but also on the quality of implementation and the psychophysiological support involved.

This multiplicity of approaches and intervening factors highlights the complex interconnection among studies involving physical activity and anxiety, as well as the influence of multiple physiological, psychological, and social factors that must be considered (65,94,98,100,104–106). It is important to emphasize that, although this review focuses on test anxiety, most of the available evidence refers to general anxiety. Therefore, the extrapolation of findings should be made cautiously, acknowledging that the gap in the literature regarding specific interventions for test anxiety persists and warrants further investigation in future studies.

The subject of this review is a relatively recent field of study, especially regarding the context of test anxiety and the use of physical activity as a coping mechanism, and despite the scarcity of clinical studies, the diversity of physical activity applications demonstrates the pursuit of preventive methods to address anxiety, particularly test anxiety, and to improve students’ mental health.[…]

Comments 5: Some reference details appear incomplete or inconsistent (years, journal metadata, and links). Please verify all references and formatting (e.g., Gallego 2014 entry; Li et al. "2021/2022" inconsistency).

Response 5: Thank you for your observation. All references have been reviewed and formatted, on pages 17 to 23.

Comments 6:  Grammatical and typographic issues (e.g., comma decimals, capitalization, and using the word "we" in a scientific context is incorrect; you can replace it with" the researchers") require professional English editing.

Response 6: Thanks for bringing this to our attention. The word 'we' has been corrected on line 161: “The researchers reviewed studies that”. We submitted the article to MDPI Author Services for English editing, to improve our manuscript’s language.

Your insights were invaluable.

Thank you!

Reviewer 2 Report

Comments and Suggestions for Authors

Explain and include in the text why only articles written in English and Portuguese were considered.

Author Response

We would like to thank you and the editorial team for your time and constructive feedback, which greatly contributed to improving our manuscript.

Below, we provide detailed responses to each comment.

Sincerely,
Iransé Oliveira Silva, PhD

Detailed Responses to Reviewer 2's Comments and Suggestions

Comments 1: Explain and include in the text why only articles written in English and Portuguese were considered.

Response 1: Thank you for your observation. The language restriction encompassed studies in English to broaden coverage and understanding of relevant, up-to-date evidence, as well as the possibility of including studies in Portuguese to enhance the understanding of regional specificities and the scientific gaps affecting this population. However, no Portuguese language articles were selected; see P. 4, lines 168 to 175.

Thank you!

Reviewer 3 Report

Comments and Suggestions for Authors

The authors are to be commended for conducting a rigorous and timely systematic review addressing the important link between physical activity and test anxiety in students. The quality of the manuscript is very good, providing a valuable synthesis of the literature from 2014 to 2024.

The manuscript demonstrates an excellent application of systematic review methodology. Above all, I appreciated the use of methodological approaches such as PICOS to define inclusion criteria, the adherence to the PRISMA statement for reporting, and the standardized assessment of methodological quality using the PEDro scale and bias risk using the RoB 2.0 tool; these instruments improve the reliability of the findings.

The overall reporting is clear; however, a minor point regarding the screening procedure in the "Materials and Methods" section requires clarification.

In the "Selection and data extraction procedure" section (lines 183-190), the autors state:

"Two researchers independently conducted the screening, and in cases of discrepancies, a third researcher was consulted. The selected studies included: author, year, and reference; the number in the experimental and control groups; the age; the intervention, duration, and total number (n); PA and anxiety (effects); anxiety measures and results (Table I)."

Further, in the "Results" section (lines 204-207), the flow of articles is detailed as:

"A total of 537 articles were selected, with 54 duplicates removed, 426 excluded after reading titles and abstracts, 43 excluded for not meeting inclusion and exclusion criteria, and 14 selected after full-text review by the authors (Figure 1)."

However, when examining the PRISMA Flow Diagram (Figure 1), the graphic indicates that the initial 426 records were excluded after screening of titles and abstracts using an Automated Tool.

While the final number of included articles and the discussion aren’t disputed, the textual description that "Two researchers independently conducted the screening" seems to contradict the visual information in Figure 1, which suggests that the bulk of the initial screening (426 exclusions) was handled by an Automated Tool.

Please clarify in the "Materials and Methods" section how the initial screening of the large number of records was executed (i.e., whether the automated tool was a first-pass step, followed by independent screening by two researchers, or if the initial screening was entirely manual). This clarification will ensure perfect alignment between the text and the PRISMA diagram.

I kindly ask the authors to explain and, if necessary, correct the following observation.

Author Response

We would like to thank you and the editorial team for your time and constructive feedback, which greatly contributed to improving our manuscript.

Below, we provide detailed responses to each comment.

Sincerely,
Iransé Oliveira Silva, PhD

Detailed Responses to Reviewer 3's Comments and Suggestions

Comments 1: The authors are to be commended for conducting a rigorous and timely systematic review addressing the important link between physical activity and test anxiety in students. The quality of the manuscript is very good, providing a valuable synthesis of the literature from 2014 to 2024.

Response 1: Thank you so much.

Comments 2: The manuscript demonstrates an excellent application of systematic review methodology. Above all, I appreciated the use of methodological approaches such as PICOS to define inclusion criteria, the adherence to the PRISMA statement for reporting, and the standardized assessment of methodological quality using the PEDro scale and bias risk using the RoB 2.0 tool; these instruments improve the reliability of the findings.

Response 2: Thank you.

Comments 3: The overall reporting is clear; however, a minor point regarding the screening procedure in the "Materials and Methods" section requires clarification.

In the "Selection and data extraction procedure" section (lines 183-190), the autors state:

"Two researchers independently conducted the screening, and in cases of discrepancies, a third researcher was consulted. The selected studies included: author, year, and reference; the number in the experimental and control groups; the age; the intervention, duration, and total number (n); PA and anxiety (effects); anxiety measures and results (Table I)."

Further, in the "Results" section (lines 204-207), the flow of articles is detailed as:

"A total of 537 articles were selected, with 54 duplicates removed, 426 excluded after reading titles and abstracts, 43 excluded for not meeting inclusion and exclusion criteria, and 14 selected after full-text review by the authors (Figure 1)."

However, when examining the PRISMA Flow Diagram (Figure 1), the graphic indicates that the initial 426 records were excluded after screening of titles and abstracts using an Automated Tool.

While the final number of included articles and the discussion aren’t disputed, the textual description that "Two researchers independently conducted the screening" seems to contradict the visual information in Figure 1, which suggests that the bulk of the initial screening (426 exclusions) was handled by an Automated Tool.

Please clarify in the "Materials and Methods" section how the initial screening of the large number of records was executed (i.e., whether the automated tool was a first-pass step, followed by independent screening by two researchers, or if the initial screening was entirely manual). This clarification will ensure perfect alignment between the text and the PRISMA diagram.

I kindly ask the authors to explain and, if necessary, correct the following observation.

Response 3: We appreciate your input. The figure and text have been corrected with the appropriate values on page 5, lines 214-217.

P5… “A total of 537 articles were identified, and 54 duplicates were removed during the identification phase. In the screening phase, 426 were excluded after reviewing titles and abstracts, and 43 were excluded for not meeting the inclusion criteria; finally, 14 were selected after a full-text review by the authors (Figure 1)”.

Text regarding heterogeneity (interventions, outcomes, populations) has been added on page 16, lines 433 to 479.

[…]Despite the observations pointed out, it is worth noting that the limitations of the studies regarding general anxiety, due to heterogeneity, do not allow for a more robust synthesis of evidence. The variability is evident in the nature and intensity of the interventions, ranging from qigong Baduanjin practice (82) and breathing exercises to high-intensity interval training (HIIT) (79) and digital interventions promoting physical activity (80,81). Session frequency and duration also varied widely, with durations from 10 minutes (81) and interventions up to 12 weeks (82), affecting the magnitude and persistence of the effects. Regarding application format, face-to-face and supervised approaches (79,80,82) and technology-mediated interventions (81) were implemented. The participants’ characteristics are equally diverse, including female university students (82), homeschooled youth (81), general university students (79), and young people with mild to moderate symptoms of depression and anxiety (80), each group presenting distinct potential moderating factors. Finally, the diversity of measurement instruments used, such as SCL90 (82), SCAS (81), STAI (79), and BAI (80), contributes to results that are not directly comparable, since each tool has varied sensitivity and specificity in capturing the various dimensions of the anxiety construct.

In studies on test anxiety, heterogeneity-related limitations include the nature of the intervention, which ranges from respiratory biofeedback (86) and mindful breathing (89) to pranayama techniques (88). Dosage also differs significantly, with daily practices lasting a few minutes to extended weekly sessions, over periods ranging from weeks to months (86,88,89). The application format includes autonomous interventions with remote support (86,89) as well as implementations integrated into supervised classroom environments (88). The participants are mostly university students, although with different anxiety profiles and contexts (86,88,89). Finally, the measurement instruments are varied, each with its own specificity for assessing test anxiety and related constructs (TAI, DASS-21, WHO-QOL-BREF, RTA, ATQ-P, PANAS, ELLAS, FLTAS).

The heterogeneity observed among the studies reflects the multifactorial nature of the relationship between physical activity and anxiety, marked by different intensities, durations, modalities, and degrees of supervision. The results indicate that structured and supervised protocols, especially those based on biofeedback, mindful breathing, and resistance training, present greater effect magnitudes, while self-directed or unsupervised interventions show minimal impact. This variability suggests that the effectiveness of physical activity on anxiety depends not only on the type of exercise but also on the quality of implementation and the psychophysiological support involved.

This multiplicity of approaches and intervening factors highlights the complex interconnection among studies involving physical activity and anxiety, as well as the influence of multiple physiological, psychological, and social factors that must be considered (65,94,98,100,104–106). It is important to emphasize that, although this review focuses on test anxiety, most of the available evidence refers to general anxiety. Therefore, the extrapolation of findings should be made cautiously, acknowledging that the gap in the literature regarding specific interventions for test anxiety persists and warrants further investigation in future studies.

The subject of this review is a relatively recent field of study, especially regarding the context of test anxiety and the use of physical activity as a coping mechanism, and despite the scarcity of clinical studies, the diversity of physical activity applications demonstrates the pursuit of preventive methods to address anxiety, particularly test anxiety, and to improve students’ mental health.[…]

We are immensely grateful for your time and dedication in improving our article. We have learned a great deal from your comments and suggestions, which have contributed significantly to improving the clarity and quality of our manuscript.

Thank you!

Reviewer 4 Report

Comments and Suggestions for Authors

Thank you for tackling a timely topic. However, the manuscript needs major revision in methodology, reporting, and language before it can be considered.

  1. Scope and conceptual focus: he introduction and conclusions frequently generalize from trials that assess general anxiety or stress, not test anxiety per se. Please define test anxiety clearly (e.g., construct, typical instruments), justify why it warrants a separate synthesis, and limit inclusion and interpretation to studies that actually measure test-anxiety outcomes. Replace broad, generic claims about anxiety with concise, recent, and directly relevant sources on test anxiety in educational settings. Avoid redundancy and ensure every paragraph advances the argument that motivates the review.
  2. Methods-transparency: State explicit PICO(S): population (age range, educational level), intervention (type/intensity/dose of physical activity), comparator, outcomes (test anxiety instruments), study design (if you restrict to RCTs), language limits, and time window. Explain and justify each criterion. Ensure all inclusions comply; otherwise, remove or justify deviations in sensitivity analyses. Provide complete, database-specific search strings (Boolean logic, field tags, truncation), the exact date of the last search, and all information sources. If you imposed language limits, justify them and discuss potential bias. Describe the process in detail: number and training of reviewers, independent screening at each stage, calibration exercises, inter-rater agreement, and conflict resolution. Use Cochrane RoB 2 correctly (cite and attribute properly). Report domain-level judgments for each trial, rationale for judgments, and visual summaries.
  3. Study selection:  Identify clearly which trials measured test anxiety as a primary outcome, which as secondary, and which measured only general anxiety.
  4. Analysis: Report actual effect estimates (e.g., mean differences or standardized mean differences) with confidence intervals and p-values where available, not only narrative statements. Distinguish statistically non-significant from clinically unimportant effects and avoid language that implies benefit without corresponding data.
  5. Discussion: Discuss plausible reasons for variability (intervention type/intensity, session dosage, delivery format, participant characteristics, outcome instruments, timing). Note construct differences between test-anxiety measures and general anxiety scales. Expand limitations (study quality, outcome inconsistency, potential publication/language bias, small samples). Translate findings into cautious, context-specific implications for educators and clinicians without overreach. If evidence is mainly for general anxiety, say so plainly. 
Comments on the Quality of English Language

The manuscript mixes English with Portuguese/Spanish fragments and contains recurring grammatical, punctuation, and typographic issues (e.g., decimal commas). A thorough language edit will improve clarity and credibility.

Author Response

We would like to thank you and the editorial team for your time and constructive feedback, which greatly contributed to improving our manuscript.

Below, we provide detailed responses to each comment.

Sincerely,
Iransé Oliveira Silva, PhD

Detailed Responses to Reviewer 4's Comments and Suggestions

Comments 1: Thank you for tackling a timely topic. However, the manuscript needs major revision in methodology, reporting, and language before it can be considered.

Response 1: We appreciate your valuable time and collaboration in improving this systematic review. We have completed all requested revisions and submitted the article to MDPI Author Services for English editing, aiming to improve our manuscript’s language.

Comments 2: Scope and conceptual focus: The introduction and conclusions frequently generalize from trials that assess general anxiety or stress, not test anxiety per se. Please define test anxiety clearly (e.g., construct, typical instruments), justify why it warrants a separate synthesis, and limit inclusion and interpretation to studies that actually measure test-anxiety outcomes. Replace broad, generic claims about anxiety with concise, recent, and directly relevant sources on test anxiety in educational settings. Avoid redundancy and ensure every paragraph advances the argument that motivates the review.

Response 2: We appreciate the reviewer's fundamental and precise observation. We recognize the importance of delineating the scope of our work specifically to test anxiety, distinguishing it from broader constructions such as general anxiety or stress, and we fully agree on the necessity of a rigorous conceptual focus. We have conducted a review to address all suggestions.

The introductory section, p. 2, lines 48 to 71, presents a definition of test anxiety as a distinct psychological construct. We highlight the presence of specific intrusive thoughts, such as the fear of failing or not meeting expectations (cognitive dimension), as well as physical symptoms such as sweating, tachycardia, and muscle tension, which are related to the affective dimension termed emotionality and directly interfere with academic/professional performance; these features are less common in other types of anxiety.

P2 … “Test anxiety is a specific form of anxiety that occurs in evaluation or testing situations, involving psychological, physiological, and behavioral reactions associated with concerns about outcomes, and is understood as synonymous with the fear of failure. It generates tension and worry (1,2). It is a response of the organism to stress, extensively studied since 1950, and negatively affects academic performance as well as students' mental health (1–6). In the evaluation process, tests have been widely used in educational environments (7,8), where test anxiety can occur before, during, or after assessments, impacting students' lives (1,2,9,10). Students may have difficulty interpreting, evaluating, and organizing their thoughts to respond to test items (11).”

The distinctiveness of test anxiety compared with other forms of anxiety (e.g., generalized or social) was detailed and extensively emphasized, focusing on its affective and cognitive dimensions. As the Introduction already indicates, physical symptoms characterize 'emotionality,' whereas 'specific intrusive thoughts, such as the fear of failing or not meeting expectations' constitute the 'worry' dimension; these are less common in other types of anxiety and directly interfere with academic or professional performance.

P2 … “Although it shares some characteristics with other types of anxiety, such as generalized or social anxiety, which can occur in a variety of everyday contexts, test anxiety has particularities that differentiate it (12). Physical symptoms such as sweating, tachycardia, and muscle tension are related to the affective dimension known as emotionality, which refers to physiological arousal. In addition to these, students experience specific intrusive thoughts, such as the fear of failing or not meeting expectations, a cognitive dimension known as worry (13–15). These thoughts are less common in other types of anxiety and directly interfere with academic or professional performance (15). Students experiencing this type of anxiety may have difficulties concentrating or recalling information during assessments, as well as a tendency to procrastinate in the weeks and months leading up to exams, limiting their study capacity (13,16). Although both types of anxiety share risk factors, such as genetic predisposition and life experiences, test anxiety may be more related to social pressures and performance expectations in educational environments (1,13,17,18). Test anxiety is a specific form of anxiety that presents unique characteristics compared to other forms, such as generalized or social anxiety (15).”

The rationale for a separate synthesis is presented on p. 2, lines 72 to 87, based on the specificity of its impact on educational outcomes and students’ mental health. We added a paragraph for better understanding on lines 96 to102.

P2… “Test anxiety has consistently shown a negative relationship with educational outcomes, where tests have taken on a much more prominent role in a multitude of important educational decisions (12). Currently, there is much greater pressure for results, and many students feel overwhelmed by stress (19). It is noted that since 2000, there has been a growing prevalence of mental illnesses, including test anxiety, partly due to increased attention to overall mental health, with more refined diagnoses and greater treatment options, and partly due to a real increase in the number of affected youths (19). Anxiety and depression have risen among university students, especially after the outbreak of the coronavirus disease 2019 (COVID-19) (20) The pandemic has led to complex situations of social isolation and prolonged screen exposure among adolescents (21), while physical activity has shown promise in addressing mental health issues, with regular physical activity aiding in anxiety reduction and improving psychological well-being (22,23). However, the specific relationship between physical activity and test anxiety is less explored, especially in this context, where stress related to academic performance may have intensified in recent years. Thus, it became necessary to analyze this relationship during the period from 2014 to 2024.”

P2 e P3… “In the school environment, little attention is given to anxiety related to examinations; however, both academic performance and subjective well-being play a fundamental role in people’s lives (30,31). Therefore, interventions targeting this type of anxiety can serve as a practical strategy to guide preventive efforts (32), since anxiety affects students’ evaluation of their own skills and achievements (33), making this study relevant for understanding and addressing this phenomenon, especially during adolescence, a period characterized by intense cognitive, emotional, and social changes (34).”

Comments 3: Methods-transparency: State explicit PICO(S): population (age range, educational level), intervention (type/intensity/dose of physical activity), comparator, outcomes (test anxiety instruments), study design (if you restrict to RCTs), language limits, and time window. Explain and justify each criterion. Ensure all inclusions comply; otherwise, remove or justify deviations in sensitivity analyses. Provide complete database-specific search strings (Boolean logic, field tags, truncation), the exact date of the last search, and all information sources. If you imposed language limits, justify them and discuss potential bias. Describe the process in detail: number and training of reviewers, independent screening at each stage, calibration exercises, inter-rater agreement, and conflict resolution. Use Cochrane RoB 2 correctly (cite and attribute properly). Report domain-level judgments for each trial, rationale for judgments, and visual summaries.

Response 3: We appreciate your input. With the aim of analyzing the relationship and effects of physical activity in managing anxiety, especially test anxiety, the inclusion criteria were reassessed to ensure clarity regarding the student population and educational level; physical activity was defined as any activity involving energy expenditure undertaken with the intent to benefit students’ mental health; the inclusion of studies that allowed between-group comparisons; and the inclusion of studies that explicitly assessed and reported general anxiety and test anxiety as outcomes, from 2014 to 2024, since earlier reviews had already addressed the topic in preceding years. The language restriction encompassed studies in English to broaden coverage and understanding of relevant, up-to-date evidence, as well as the possibility of including studies in Portuguese to enhance the understanding of regional specificities and the scientific gaps affecting this population. However, no Portuguese language articles were selected. P. 4, lines 168 to 175.

P4 … “For inclusion criteria, the PICOS approach was adopted (72). Studies were selected with students aged 13 to 25 years as the population, covering educational stages from secondary to higher education and equivalent qualifications; physical activity (PA) or exercise as intervention; a comparison between those who participated and those who did not; and anxiety, specifically test anxiety and mental health, as outcomes. Randomized and controlled trials, cluster-randomized trials and quasi-experimental studies published between January 2014, and January 2024 were included. Peer-reviewed publications writ-ten in English or Portuguese, regardless of the country, were considered.”

As for the compliance of the included studies, this systematic review focuses on exercise to minimize stress prior to assessments among students. According to the Template for Intervention Description and Replication (TIDieR), there is no specific recommendation regarding blinding in clinical trials of exercise-based interventions. The TIDieR guidelines aim to enhance the quality of intervention reporting, particularly dosage parameters such as intensity and frequency, to make intervention protocols replicable in studies such as randomized controlled trials (RCTs). This is believed to improve publication transparency, facilitate evidence synthesis, and support implementation in clinical practice (Hoffmann et al., 2014, https://doi.org/10.1136/bmj.g1687).

The studies included in this review received lower scores precisely on items 5 to 7 of the PEDro scale, which assess blinding. On the other items, they were rated as moderate-to-high quality, which helps justify retaining the studies in the review and minimizes potential biases in sensitivity analyses. Moreover, the specificity of the topic should be considered, given that the exercise protocols included in the studies were planned and systematized for the stress context associated with curricular examinations.

It is worth emphasizing the high clinical relevance of the included studies, as exercise interventions mitigate stress-related mental health problems in this age group and reduce public healthcare expenditures. Finally, in the exploratory RoB analysis, some studies showed a high risk of bias with respect to the randomization process, which includes blinding. Thus, we underscore that most of the included studies employed physical exercise protocols, which in many cases do not necessitate or feasibly permit blinding.

For the selection of articles, we conducted a search across databases combining the following MeSH terms, pp. 180 to 86.

Datebase

Total

Code

PubMed

188

(("Exercise"[MeSH] OR "Physical Activity"[MeSH] OR "Motor Activity"[MeSH] OR "Exercise Therapy"[MeSH] OR "exercise"[Title/Abstract] OR "exercises"[Title/Abstract] OR "physical activity"[Title/Abstract] OR "physical exercise"[Title/Abstract] OR "training"[Title/Abstract]) AND ( ("Anxiety"[MeSH] OR "Test Anxiety"[MeSH] OR "anxiety"[Title/Abstract] OR "test anxiety"[Title/Abstract] OR "exam stress"[Title/Abstract] OR "academic stress"[Title/Abstract] OR "performance anxiety"[Title/Abstract])) AND (("Mental Health"[MeSH] OR "Stress, Psychological"[MeSH] OR "Psychological Well-Being"[MeSH] OR "mental health"[Title/Abstract] OR "psychological stress"[Title/Abstract] OR "emotional well-being"[Title/Abstract] OR "psychological distress"[Title/Abstract])) AND (("Students"[MeSH] OR "Adolescent"[MeSH] OR "Young Adult"[MeSH] OR "Schools"[MeSH] OR "students"[Title/Abstract] OR "adolescents"[Title/Abstract] OR "youth"[Title/Abstract] OR "school students"[Title/Abstract] OR "college students"[Title/Abstract]))

Web of Science

46

TS=(("exercise" OR "exercises" OR "physical activity" OR "physical exercise" OR "training")AND("anxiety" OR "test anxiety" OR "exam stress" OR "academic stress" OR "performance anxiety")AND("mental health" OR "psychological stress" OR "psychological distress" OR "emotional well-being")AND("students" OR "adolescents" OR "youth" OR "school students" OR "college students"))

Lilacs

55

(("Exercise"[mh] OR "Physical Activity"[mh] OR "Motor Activity"[mh] OR "Exercise Therapy"[mh] OR "exercise"[tiab] OR "exercises"[tiab] OR "physical activity"[tiab] OR "physical exercise"[tiab] OR "training"[tiab])AND("Anxiety"[mh] OR "Test Anxiety"[mh] OR "anxiety"[tiab] OR "test anxiety"[tiab] OR "exam stress"[tiab] OR "academic stress"[tiab] OR "performance anxiety"[tiab])AND("Mental Health"[mh] OR "Stress, Psychological"[mh] OR "Psychological Well-Being"[mh] OR "mental health"[tiab] OR "psychological stress"[tiab] OR "emotional well-being"[tiab] OR "psychological distress"[tiab])AND("Students"[mh] OR "Adolescent"[mh] OR "Young Adult"[mh] OR "Schools"[mh] OR "students"[tiab] OR "adolescents"[tiab] OR "youth"[tiab] OR "school students"[tiab] OR "college students"[tiab]))

Cochrane

14

(("Exercise":ti,ab,kw OR "Physical Activity":ti,ab,kw OR "Motor Activity":ti,ab,kw OR "Exercise Therapy":ti,ab,kw OR "exercise":ti,ab,kw OR "exercises":ti,ab,kw OR "physical activity":ti,ab,kw OR "physical exercise":ti,ab,kw OR "training":ti,ab,kw)AND("Anxiety":ti,ab,kw OR "Test Anxiety":ti,ab,kw OR "anxiety":ti,ab,kw OR "test anxiety":ti,ab,kw OR "exam stress":ti,ab,kw OR "academic stress":ti,ab,kw OR "performance anxiety":ti,ab,kw)AND("Mental Health":ti,ab,kw OR "Stress, Psychological":ti,ab,kw OR "Psychological Well-Being":ti,ab,kw OR "mental health":ti,ab,kw OR "psychological stress":ti,ab,kw OR "emotional well-being":ti,ab,kw OR "psychological distress":ti,ab,kw)AND("Students":ti,ab,kw OR "Adolescent":ti,ab,kw OR "Young Adult":ti,ab,kw OR "Schools":ti,ab,kw OR "students":ti,ab,kw OR "adolescents":ti,ab,kw OR "youth":ti,ab,kw OR "school students":ti,ab,kw OR "college students":ti,ab,kw))

Medline

234

((Exercise/ OR Physical Activity/ OR Motor Activity/ OR Exercise Therapy/ OR exercise.mp. OR exercises.mp. OR physical activity.mp. OR physical exercise.mp. OR training.mp.)AND(Anxiety/ OR Test Anxiety/ OR anxiety.mp. OR test anxiety.mp. OR exam stress.mp. OR academic stress.mp. OR performance anxiety.mp.)AND(Mental Health/ OR Stress, Psychological/ OR Psychological Well-Being/ OR mental health.mp. OR psychological stress.mp. OR emotional well-being.mp. OR psychological distress.mp.)AND(Students/ OR Adolescent/ OR Young Adult/ OR Schools/ OR students.mp. OR adolescents.mp. OR youth.mp. OR school students.mp. OR college students.mp.))

Date of last search

We used Rayyan to manage study selection: records were imported, screening was performed under blinding, uncertainties were resolved in consensus meetings, the final set of included studies was determined, and data were organized in Microsoft Excel. We then calibrated the PEDro scale and risk-of-bias procedures in line with the Revised Cochrane risk-of-bias tool for randomized trials (RoB 2) guidance and the Revised Cochrane risk-of-bias tool for cluster-randomized trials (RoB 2 CRT) Template for Completion (20210318_RoB_2_guidance_cluster_trial .pdf - Google Drive) and Revised Cochrane risk-of-bias tool for cluster-randomized trials (RoB 2 CRT) TEMPLATE FOR COMPLETION (20210318_RoB_2_template_cluster_trial.docx - Google Drive). After completing the assessments, we resolved discrepancies and generated the visualizations in the software, opting for the summary (weighted bar) plot rather than the traffic-light diagram.

Comments 4: Study selection: Identify clearly which trials measured test anxiety as a primary outcome, which is secondary, and which measured only general anxiety.

Response 4: Thanks for bringing this to our attention. A column has been included in Table I identifying the types of outcomes, on pages 6 to 10.

Comments 5: Analysis: Report actual effect estimates (e.g., mean differences or standardized mean differences) with confidence intervals and p-values where available, not only narrative statements. Distinguishing statistically non-significant from clinically unimportant effects and avoiding language that implies benefit without corresponding data.

Response 5: Thank you. We reviewed Table I, identifying means, standard deviations, and p-values, as well as checking effect estimates and conducting sensitivity analysis of the studies. We have added explanatory text on page 5, lines 219-221, and also on page 12, lines 237-234.

P12... Although 64.28% of the 14 selected studies demonstrated a reduction in anxiety through the use of physical activity (PA) modalities, such as aerobic exercise, mind-fulness, games, relaxation, walking, Baduanjin (Qigong), conscious and slow breathing, self-directed breathing, biofeedback, pranayamic breathing, and resistance training the analysis of the selected studies revealed wide variability in the magnitude of the effects of physical activity and psychophysiological interventions on anxiety. The effect sizes ranged from 0.005 to 1.543, covering null to very large effects.

            Resistance training showed the largest effect (RT = 1.543) (57), followed by biofeedback combined with psychoeducation (BIOF = 0.820) (86), mindful breathing practice (MBP = 0.780), and cognitive reappraisal (CRP = 0.626) (89). Smaller magnitudes were observed for mindfulness (MBCT = 0.342) and Physical Education practice (PE = 0.161) (87).

            Self-directed interventions, such as autonomous breathing and pranayama, presented small or negligible effects (0.005–0.141) (84,85,88). Overall, the findings indicate that structured and supervised protocols tend to produce more robust effects compared to unsupervised approaches. It is noteworthy that only 14.28% of the studies incorporated complementary measures such as heart rate variability (HRV) and biological and/or physiological analyses to confirm the effectiveness of physical activity.

Comments 6: Discussion: Discuss plausible reasons for variability (intervention type/intensity, session dosage, delivery format, participant characteristics, outcome instruments, timing). Note construct differences between test-anxiety measures and general anxiety scales. Expand limitations (study quality, outcome inconsistency, potential publication/language bias, small samples). Translate findings into cautious, context-specific implications for educators and clinicians without overreach. If evidence is mainly for general anxiety, say so plainly.

Response 6: Your insight is invaluable. We acknowledge the importance of discussing the variability observed in the included studies, as well as the limitations associated with the current evidence, and we have revised the Discussion, on page 16, lines 433 to 480.

Despite the perceptions pointed out, it is worth noting that the limitations of the studies regarding general anxiety, due to heterogeneity, do not allow for a more robust synthesis of evidence. The variability is evident in the nature and intensity of the interventions, ranging from Qigong Baduanjin practice(82) and breathing exercises to High-Intensity Interval Training (HIIT) (79) and digital interventions promoting physical activity (80,81). Session dosage also varies widely, with durations ranging from 10 minutes (81) to 12 weeks of intervention (82), affecting the magnitude and persistence of the effects. Regarding the application format, both face-to-face and supervised approaches (79,80,82) and technology-mediated interventions (81) are observed. Participant characteristics are equally diverse, including female university students (82), homeschooled youth (81), general university students (79) and young people with mild to moderate symptoms of depression and anxiety (80), each group presenting distinct potential moderating factors. Finally, the diversity of measurement instruments used, such as SCL90 (82), SCAS (81), STAI (79), and BAI (80), contributes to results that are not directly comparable, since each tool has different sensitivity and specificity to capture the various dimensions of the anxiety construct.

In studies on test anxiety, heterogeneity-related limitations include the nature of the intervention, which ranges from respiratory biofeedback (86) and mindful breathing (89) to pranayama techniques (88). Dosage also differs significantly, with daily practices lasting a few minutes to extended weekly sessions, over periods ranging from weeks to months (86,88,89). The application format includes autonomous interventions with remote support (86,89) as well as implementations integrated into supervised classroom environments (88). Participant characteristics are mostly focused on university students, although with different anxiety profiles and contexts (86,88,89). Finally, the measurement instruments are varied, each with its specificity for assessing test anxiety and related constructs (TAI, DASS-21, WHO-QOL-BREF, RTA, ATQ-P, PANAS, ELLAS, FLTAS).

The heterogeneity observed among the studies reflects the multifactorial nature of the relationship between physical activity and anxiety, marked by different intensities, durations, modalities, and degrees of supervision. The results indicate that structured and supervised protocols, especially those based on biofeedback, mindful breathing, and resistance training, present greater effect magnitudes, while self-directed or unsupervised interventions show minimal impact. This variability suggests that the effectiveness of physical activity on anxiety depends not only on the type of exercise but also on the quality of implementation and the psychophysiological support involved.

This multiplicity of approaches and intervening factors highlights the complex interconnection among studies involving physical activity and anxiety, as well as the influence of multiple physiological, psychological, and social factors that must be considered (65,94,98,100,104–106). It is important to emphasize that, although this review focuses on test anxiety, most of the available evidence refers to general anxiety. Therefore, the extrapolation of findings should be made cautiously, acknowledging that the gap in the literature regarding specific interventions for test anxiety persists and deserves further investigation in future studies.

It is a relatively recent field of study, especially regarding the context of test anxiety and the use of physical activity as a coping mechanism, and despite the scarcity of clinical studies, the diversity of physical activity applications demonstrates the pursuit of preventive methods to address anxiety, particularly test anxiety, and to improve students’ mental health.

Comments 7: Comments on the Quality of English Language The manuscript mixes English with Portuguese/Spanish fragments and contains recurring grammatical, punctuation, and typographic issues (e.g., decimal commas). A thorough language edit will improve clarity and credibility.

Response 7: Thank you for your observations. We have reviewed the manuscript, and the manuscript has undergone professional English language and grammar editing.

We are immensely grateful for your time and dedication in improving our article. We have learned a great deal from your comments and suggestions, which have contributed significantly to improving the clarity and quality of our manuscript.

Thank you!

Round 2

Reviewer 4 Report

Comments and Suggestions for Authors

The manuscript presents a clearly improved version compared to the previous one. The authors have addressed most of the comments made in the first round of review carefully and systematically, introducing changes that have helped to clarify the structure of the text, strengthen the theoretical framework, and improve the presentation and discussion of the results.

I would like to sincerely thank the authors for the effort, time, and dedication invested in revising the manuscript, incorporating new references, refining some statements, and clarifying the methodology used. These modifications contribute significantly to the scientific quality and clarity of the article, facilitating its understanding and increasing its potential interest for the academic community.

Overall, I consider that the work has progressed substantially and that the revisions made reflect a constructive and open attitude on the part of the authors, which is highly positive within the peer-review process.